# Quantitative Risk Evaluation of Antimicrobial-Resistant *Vibrio parahaemolyticus* Isolated from Farmed Grey Mullets in Singapore

**DOI:** 10.3390/pathogens12010093

**Published:** 2023-01-05

**Authors:** Hong Ming Glendon Ong, Yang Zhong, Cheng Cheng Hu, Kar Hui Ong, Wei Ching Khor, Joergen Schlundt, Kyaw Thu Aung

**Affiliations:** 1School of Chemistry, Chemical Engineering and Biotechnology, Nanyang Technological University, Block N1.2, B3-15, 62 Nanyang Drive, Singapore 637459, Singapore; 2National Centre for Food Science, Singapore Food Agency, 7 International Business Park, Techquest, Singapore 609919, Singapore; 3Department of Clinical Translational Research, Singapore General Hospital, Academia, 20 College Road, Singapore 169856, Singapore; 4Singapore Institute of Manufacturing Technology, 2 Fusionopolis Way, 08-04, Innovis, Singapore 138634, Singapore; 5Schlundt Consult, 3250 Gilleleje, Denmark; 6School of Biological Sciences, Nanyang Technological University, Singapore 637551, Singapore

**Keywords:** antimicrobial-resistance risk assessment, haemolytic, @Risk software, farm-to-home, retail-to-home, sensitivity analysis

## Abstract

*Vibrio parahaemolyticus* is a causative pathogen for gastroenteritis involving the consumption of undercooked or raw seafood. However, there is a paucity of data regarding the quantitative detection of this pathogen in finfish, while no study reported the enumeration of haemolytic antimicrobial-resistant (AMR) *V. parahaemolyticus*. In this study, ampicillin-, penicillin G- and tetracycline-resistant and non-AMR haemolytic *V. parahaemolyticus* isolates were monitored and quantified in grey mullet samples reared locally from different premises within the food chain (farm and retail). Occurrence data for haemolytic *V. parahaemolyticus* were 13/45 (29%) in farm fish samples, 2/6 (one third) from farm water samples and 27/45 (60%) from retail fish samples. Microbial loads for haemolytic *V. parahaemolyticus* microbial loads ranged from 1.9 to 4.1 Log CFU/g in fish samples and 2.0 to 3.0 Log CFU/g in farm water samples. AMR risk assessments (ARRAs) for both the full farm-to-home and partial retail-to-home chains in the risk modelling framework were conducted, specifically for ampicillin, penicillin G, tetracycline and haemolytic (non-AMR) scenarios. The haemolytic ARRA predicted an average probability of illness of 2.9 × 10^−4^ and 4.5 × 10^−5^ per serving for the farm-to-home and retail-to-home chains, respectively, translating to 57 and 148 cases annually. The ratios of the average probability of illness per year for the three ARRAs to the haemolytic ARRA were 1.1 × 10^−2^ and 3.0 × 10^−4^ (ampicillin and penicillin G, respectively) for the farm-to-home chain and 1.3, 1.6 and 0.4 (ampicillin, penicillin G and tetracycline, respectively) for the retail-to-home chain. Sensitivity analysis showed that the initial concentrations of haemolytic *V. parahaemolyticus* in the gills and intestines of the fish and the cooking and washing of the fish cavity were the major variables influencing risk outputs in all modelled ARRAs. The findings of this study are useful for relevant stakeholders to make informed decisions regarding risk management to improve overall food safety.

## 1. Introduction

*Vibrio parahaemolyticus* is found ubiquitously in estuarine and marine environments and proliferates extensively in warmer seasons. As a foodborne pathogen, *V. parahaemolyticus* has been known to cause acute gastroenteritis, characterised by abdominal cramps, diarrhoea, vomiting, nausea and fever, as well as wound infection and septicaemia [1]). In the United States, *V. parahaemolyticus* is the leading cause of gastroenteritis associated with the consumption of undercooked or raw seafood, with an estimated 35,000 cases yearly [2]. However, in Singapore, gastroenteritis caused by *V. parahaemolyticus* is not within the list of infectious diseases legally notifiable by law. Hence, there is a need to conduct quantitative microbial risk assessments to assess the incidences and impacts caused by this specific pathogen in the food chain.

Studies have shown that not all *V. parahaemolyticus* isolates have been known to cause disease. While *V. parahaemolyticus* pathogenicity is complex, virulent clinical strains isolated from patients with gastroenteritis have been shown to exhibit the kanagawa phenomenon, wherein a beta-haemolysis pattern is observed on wagatsuma agar containing human blood, while other virulent clinical strains form a weak haemolytic zone in normal human blood agar [3,4]. The haemolytic reactions of such virulent strains were also observed on normal sheep blood agar with both primarily alpha-haemolysis and beta-haemolysis patterns [5].

The use of antimicrobials can be used for the treatment of severe infections caused by *V. parahaemolyticus*, as most *Vibrio* spp. are sensitive to most antimicrobials of significance [6]. As many *Vibrio* species are zoonoses, antimicrobials in aquaculture settings are used to not only control such bacterial infections but also to facilitate growth promotion in reared seafood species [7]. However, the excessive use of antimicrobials in aquaculture and human settings has led to the rapid emergence and evolution of antimicrobial resistance (AMR) in *Vibrio* spp. over the last few decades [7,8]. AMR develops when the bacteria is able to adapt and grow even in the presence of bacteriostatic or bactericidal antimicrobials, resulting in the loss of effectiveness of the drug [9]. In particular, high resistance rates to the penicillin class of drugs have been consistently reported in many countries for *V. parahaemolyticus* in environmental and clinical isolates [10,11,12,13,14]. It has also been reported that *Vibrio* isolates retrieved from marine fish in Singapore aquaculture showed consistent resistance to tetracycline [15], and recent studies show the emergence of tetracycline resistance in *V. parahaemolyticus* isolated from coastal waters and shrimp aquaculture in the Italian Adriatic Sea and Hangzhou, China respectively [16,17]. The emergence and spread of haemolytic AMR *V. parahaemolyticus* in the food chain for aquaculture can become a food safety and public health concern, as it can lead to increased infection rates, infection severity and frequency of antimicrobial treatment failure [18].

Currently, around 10% of local seafood is produced by Singapore’s aquaculture industry, with the majority of production arising from marine coastal farms with floating net cages along northern coasts of Singapore [19,20]. Grey mullet is among some of the high-nutritional-value marine fish species cultivated and is preferred due to its affordability and availability throughout the year [19]. The qualitative detection of *V. parahaemolyticus* has been studied in numerous finfishes, including grey mullet, red mullet, sardines, Atlantic mackerel and anchovies, through biochemical tests and selective medium [21]. Currently, there are only two studies reporting the prevalence and concentrations of this pathogen in different parts of finfish from experimental data [22,23]. Both studies observed varying levels of haemolytic *V. parahaemolyticus* contamination within the gills, intestines and skin of mackerel fish, and the results were subsequently used to conduct quantitative microbial risk assessments to quantify the associated risks involved. However, the quantitative detection and enumeration of AMR *V. parahaemolyticus* in finfishes remain limited in extent. As *V. parahaemolyticus* can pose serious food safety risks through the consumption of contaminated seafood, food safety and public health measures must be taken in order to mitigate, reduce or eliminate such risks. Thus, these measures can be guided with the aid of AMR risk assessments (ARRA) tools, which are designed specifically to identify and quantify such risks arising within the farm-to-fork chain.

However, the use of such ARRA tools tailored to the consumption of finfish is lacking worldwide, as well as in Singapore. In addition, there are knowledge gaps with regards to the detection and enumeration of haemolytic AMR *V. parahaemolyticus* in locally produced finfish, which are needed for the exposure assessment within ARRA. This study aims to fill in these gaps by determining the occurrence and concentration of haemolytic AMR *V. parahaemolyticus* in locally produced grey mullets within the farm and retail premises, as well as to conduct ARRAs specified for the consumption of grey mullets in the local context. Information from such tools can be used to estimate the burden of AMR in haemolytic *V. parahaemolyticus* on human health and to inform evidence-based food safety and public health measures.

## 2. Methodology

### 2.1. Collection of Survey Data for Exposure Assessment

A survey involving an open-cage marine coastal farm and a hypermarket was conducted to obtain the relevant data needed for the exposure assessment of the ARRA. Prior to the survey, there were limited data regarding the occurrence and concentration of *V. parahaemolyticus* isolated from grey mullets farmed in Singapore. Hence, the survey aimed to obtain a positive rate and microbial loads of *Vibrio parahaemolyticus* in finfish for comparisons with other similar literatures while also complementing other literature data within the ARRA model.

### 2.2. Sample Collection

#### 2.2.1. Sampling Period

The sample collection was conducted between November 2019 and January 2020.

#### 2.2.2. Farm and Retail Sampling

Five freshly harvested grey mullet samples per sampling week were obtained from a major marine fish farm. The collection of fish samples were carried out close to the harvesting stage of around three to seven days before harvest. Farm fish samples obtained were collected directly from sea cages. As Singapore’s aquaculture industry is relatively small compared to other countries, it was hypothesised that open-cage marine aquaculture systems within Singapore do not vary much across different farms. The sampling was conducted for three consecutive weeks, which added up to a total of fifteen freshly harvested grey mullet samples. All samples were individually transferred to a sterile sampling bag and immediately placed on ice. In addition, two water (1 L each) samples per sampling week were collected from the fish farm where the grey mullets were bred, which summed up to a total of six water samples. In addition, the farmer was interviewed on farm practices regarding antimicrobial usage during the rearing stage.

Five chilled grey mullet samples per sampling week was obtained from a major retail hypermarket ice-bed counter. Retail workers were consulted, and only fish samples that were harvested the day before were chosen. The sampling was conducted for three weeks, which added up to a total of fifteen chilled grey mullet samples. All samples were individually transferred to a sterile sampling bag and immediately placed on ice. Samples were then immediately transported to the laboratory and processed for laboratory testing on the same day. The weight of fish samples obtained from either the farm or retail premises were around 450–650 g.

#### 2.2.3. Sample Processing

Each grey mullet sample was weighed. The gills, intestines, and skins (with flesh) were excised using aseptic techniques, weighed and tested. A total of 30 gills, 30 intestines and 30 skin (classified as fish parts subsequently) samples from 15 live and 15 chilled grey mullet samples were analysed for laboratory testing in this study.

#### 2.2.4. Direct Plate Count of Presumptive Vibrio Species

For each fish part sample, a ten-fold dilution was carried out by transferring 9 parts of sterile 3% saline to 1 part of sample in a sterile stomacher bag. Samples were then homogenised for 90 s using a stomacher Lab-blender 400 (Seward Medical, UK). Each 1 litre water sample was filtered through a glass filtration system, and bacteria were collected using 0.45 µM nitrocellulose membrane filter (Sigma, Germany). The filter was then transferred to a tube containing 10 mL of sterile 3% saline and vortexed for 5 min to allow the bacteria to transfer to saline water. Serial dilution was then carried out for all fish homogenate samples and filtrate samples by up to 10^6^ dilution by aliquoting 0.1 mL of suspension mixture to 0.9 mL of 3% saline. The 0.1 mL of suspension mixture at each dilution factor was then spread plated on three different types of thiosulfate citrate bile salts sucrose (TCBS) agar, containing either 32 µg/mL ampicillin, 32 µg/mL penicillin G or 16 µg/mL tetracycline. Furthermore, 0.1 mL of suspension mixture at each dilution factor was also spread plated on TCBS agar that was unsupplemented. Duplicates were carried out for each sample at each dilution level for all four treatment types. The TCBS plates were incubated at 37 °C for 24 h and observed for the formation of green colonies. Plates with countable colonies were then colony-lifted onto tryptone soya agar with 5% sheep blood (Thermofisher Scientific, Waltham, MA, USA using a replica-plating tool and sterile velveteen sheets (VWR, Radnor, PA, USA) for the phenotypic screening of haemolysis. All blood agar plates were incubated at 37 °C for 24 h and observed for haemolysis patterns. Colonies that exhibited either alpha or beta haemolysis were counted and determined visually for haemolytic population.

#### 2.2.5. Phenotypic Identification of Haemolytic Strains

Several haemolytic and non-haemolytic colonies representative of the blood agar plate were then picked. Each single picked colony was then streaked on a blood agar plate and incubated at 37 °C for 24 h and observed for haemolytic activity. A single colony was then picked from the blood agar and then streaked onto Luria-Bertani Miller (LB) agar with 3% NaCl and incubated at 37 °C for 24 h. Colony morphology was observed for a purity check, and a single colony was then picked and cultured in Luria Bertani (LB) broth (Difco, Becton, NJ, USA) with 3% NaCl and incubated at 37 °C for 24 h. Glycerol stocks were then made from each pure bacterial isolate and stored at −80 °C for 16s rRNA sequencing and antimicrobial susceptibility testing.

#### 2.2.6. 16 s rRNA Gene Amplification and Sequencing

Each pure bacterial isolate was thawed, and a loopful of culture was streaked onto 3% NaCl LB agar and incubated at 37 °C for 24 h. The universal primers, forward primer 27F (5′-AGAGTTTGATCCTGGCTCAG-3′) and reverse primer 1492R (5-TACGGTTACCTTGTTACGACTT-3) were used to amplify the full length of the 16s rRNA gene. Colony PCR was carried out by gently touching 2–3 colonies and directly transferring them to a PCR reaction mixture containing 12.5 µL 2X REDiant PCR mastermix (Axil Scientific, Singapore), 1 µL 10 µM primer 27F and 1 µL 10 µM primer 1492R. The PCR was carried out on a T100 thermocycler (Biorad, United States) with the following cycling conditions: initial denaturation at 95 °C for 3 min, followed by 35 cycles of denaturation at 98 °C for 10 s, annealing at 51 °C for 15 s and extension at 72 °C for 2 min. A final extension step was performed at 72 °C for 10 min. The resulting PCR products were analysed with 1% agarose gel electrophoresis supplemented with GelRed (Sigma-Aldrich, St Louis, MO, USA). The size of the amplified 16s rRNA gene was estimated around ~1400 bp and compared with a Generuler 1kb DNA ladder (Thermofischer, Waltham, MA, USA). The PCR product was then purified for sequencing using the DNA Clean & Concentrator kit (Zymo, Irvine, CA, USA) and quantified using NanoDrop ND-100 (Thermo Scientific, Waltham, MA, USA). Purified PCR products were sent to 1st base, Axil Scientific for Sanger sequencing, which utilised the ABI-PRISM 31000 Genetic Analyzer system and BigDye Terminator v3.1 Cycle Sequencing kit chemistry. Universal primers 27F and 1492R were used to obtain forward and reverse sequences data, respectively. Forward and reverse sequences were aligned and combined using a BioEdit Sequence Alignment Editor. Taxonomic identification of the sequences from the 16s rRNA gene of the bacterial isolates were obtained using the online *BlastN* software (http://www.ncbi.nlm.nih.gov/BLAST/ (accessed on 8 February 2021)). Each bacterial isolate was compared to the top hit of the list of results with a similarity percentage ≥99.0%.

#### 2.2.7. Antimicrobial Susceptibility Testing

Antimicrobial susceptibility testing was carried out for all 271 bacterial isolates using the disc diffusion method. Each bacterial isolate was cultured in 5 mL of LB broth (BD Difco, Becton, NJ, USA) supplemented with 3% NaCl (Merck, Rahway, NJ, USA) and incubated at 37 °C for 24 h. Inocula were adjusted to 0.5 McFarland standard, and a sterile swab was dipped in the bacterial suspension and swabbed on the entire surface of Muller–Hinton agar (Oxoid, Basingstoke, HA, UK) and left to dry. Eight antimicrobial susceptibility test discs (Oxoid, UK) containing ampicillin (10 µg), ampicillin/sulbactam (10 µg/10µg), penicillin G (10 unit), tetracycline (30 µg), cefotaxime (30 µg), ciprofloxacin (5 µg), sulfamethoxazole/trimethoprim (1.25 µg/23.75µg) and chloramphenicol (30 µg) were punched on to inoculated agar plate using the disc dispenser and incubated at 37 °C for 24 h. The results were interpreted as sensitive, intermediate and resistant based on the Clinical and Laboratory Standards Institute (CLSI) M45-P guideline for *Vibrio* spp. [24]. Interpretative criteria for penicillin G not available in M45-P for *Vibrio* spp. were referred to CLSI M100 [25]. Data regarding the antimicrobial susceptibility testing of all bacterial isolates are provided in the Appendix A.

### 2.3. Quantitative Risk Evaluation Model Framework

This study aims to conduct ARRAs for ampicillin-, penicillin G-, tetracycline-resistant and non-AMR haemolytic *V. parahaemolyticus* in Singapore following the *Codex Alimentarius* guidelines for the risk analysis of foodborne pathogens carrying AMR [26]. A full farm-to-home chain and a partial retail-to-home chain ARRA were conducted to allow a comparison of results. Through the analysis of ARRA models and studied variables, intervention measures to mitigate or reduce food safety risk were recommended. The quantitative risk evaluation model framework was depicted (Figure 1). Several assumptions were made for the framework as described below:Grey mullets harvested from the farm were not processed but were quickly packed in ice and sent to the fishery port and subsequently the hypermarket. Once at the hypermarket, they were placed on open-air fish ice beds for display. Processing of the food fish were carried out at the consumer’s home.The survivability fitness of different *V. parahaemolyticus* strains was considered equal.Within the ARRA framework for a specified antimicrobial resistance, co- and cross-resistance traits of *V. parahaemolyticus* strains regarding the other two antimicrobials were not considered.Only direct exposure from the consumption of contaminated seafood to consumers was considered. Secondary transmission of infection, including transmission through workers within the food chain, was not considered.Human host immunity to infection from *V. parahaemolyticus* was not considered.

The occurrence and concentration data of haemolytic AMR *V. parahaemolyticus* genes in grey mullets at the pre-harvest stage and retail stage were obtained from the survey. The full weight of each grey mullet fish was measured to determine the concentration of haemolytic AMR *V. parahaemolyticus* for the whole fish body. Input parameters such as harvest, transport, and display duration times and temperature; consumer’s food preparatory practices; and consumption patterns and statistics were either obtained from scientific literature or surveys from relevant stakeholders. The use of local data was prioritised, and when that was impossible, surrogate data from other Southeast Asian countries were adopted.

All input variables were randomly sampled using Monte Carlo sampling with 100,000 iterations per simulation using the @RISK version 7.6 software (Pallisade Corporation) to generate output results that estimated the probability of illness from a single serving of grey mullet meal, likelihood of infection for each person per year and estimated number of cases per year of exposed population. A total of 20 simulations were performed and averages obtained. All model input parameters are summarised (Table 1).

#### 2.3.1. Hazard Identification

Haemolytic *V. parahaemolyticus* strains resistant to either penicillin G, ampicillin or tetracycline were considered microbial-agent hazards in grey mullets in this study. All strains were phenotypically tested for alpha or beta haemolysis by streaking on sheep blood agar, and strains with haemolytic activity were presumed to be clinically virulent. Strains that did not exhibit haemolytic activity were considered non-virulent and were excluded in this risk assessment.

#### 2.3.2. Exposure Assessment

ARRAs were modelled with two different scenarios within the exposure assessment phase: the full farm-to-home chain and the partial retail-to-home chain. Results from the two scenarios were compared with each other.

##### *V. parahaemolyticus* Growth Rate Modelling and Adjustment Factors

There was growth of *V. parahaemolyticus* within the grey mullets as the seafood product was harvested from the farm, transported to the hypermarket, put on display and finally purchased and transported to consumer’s home prior to preparation and cooking. The growth rate of *V. parahaemolyticus* was obtained using the broth model developed with water activity fixed at the optimum value of 0.985. Several assumptions were considered for the growth rate:The growth rate of haemolytic of *V. parahaemolyticus* were the same across all strains considered.The lag phase associated during the harvest stage was considered negligible, as there was no change in the growth environment.The growth pattern of haemolytic *V. parahaemolyticus* exhibited in grey mullet are highly similar to that in mackerel, as both share highly similar biological traits.

##### Concentration and Occurrence of Haemolytic *V. parahaemolyticus* at Pre-Harvest and Retail

Data on the concentration and occurrence of haemolytic AMR *V. parahaemolyticus* for grey mullet samples obtained at the pre-harvest stage and retail stage were obtained from the survey and split under 4 different ARRA scenarios (haemolytic, ampicillin, penicillin G and tetracycline).

##### Harvesting Conditions and Transportation to Hypermarket

Harvest temperatures and duration were modelled based on the climate report and survey with the farmer [31]. After harvest, grey mullets were packed in ice and sent to the hypermarket.

##### Retail Display and Transportation to Home

Grey mullets in the hypermarket were placed atop an open fish bed filled with crushed ice with other whole finfish products and labelled as fresh seafood. As the seafood product was directly exposed to the environment of the hypermarket and subjected to greater temperature variations, retail display temperatures were modelled with greater temperature ranges [32]. Information regarding the retail display duration were obtained by surveying workers working within the fresh seafood section within the hypermarket. The majority of the purchases (90%) were made in the morning, within the first 3.5 h, while the remaining purchases (10%) were made from 3.5 h onwards up to 12.5 h. As the grey mullet was sold as fresh seafood, it is considered perishable and will quickly turn foul when stored at high temperatures for over 1 h, rendering it unsafe for consumption [32,38]. Therefore, home-transportation duration was modelled as being under an hour.

##### Preparation, Cooking and Consumption Patterns

Preparation of the grey mullet at home includes the evisceration and removal of the fish’s internal organs and the washing of the eviscerated body cavity using clean water prior to cooking. Two different scenarios were modelled, wherein either the body cavity is washed with clean water or there is no washing of the body cavity. The washing of the body cavity has been shown to reduce microbial load within the fish, thereby reducing risk [33]. As washing is commonly practised prior to cooking, the average washing preparation was modelled using a discrete distribution of 90% and 10% for the washing and no-washing scenarios, respectively. The pan-frying method was selected for the cooking process based on heat treatment and kitchen simulation studies from Ye and Tan [32,34].

#### 2.3.3. Hazard Characterisation

##### Dose-Response Relationship

The Beta-Poisson dose-response model adopted from US FDA [29] was used to translate the exposure of haemolytic *V. parahaemolyticus* (dose) to an estimate of the probability of illness per person per serving in this study. As model parameters specific to the Singapore population are lacking, model parameters proposed by the US FDA were adopted. A non-parametric bootstrapping procedure was used to characterise the uncertainty involving the model parameters, with the probability-weighted selection of a combination of model parameters with their corresponding maximum likelihood estimates (MLEs) [29]. The combination of model parameters, along with their MLEs and probabilities, were described (Table 2).

Changes in occurrences in seafood product moving through the farm-to-retail chain was modelled.

#### 2.3.4. Risk Characterisation

The estimated number of cases per year was calculated by multiplying the probability of illness per year and the exposed population. Sensitivity analysis was carried out using @RISK software to identify the key variables that highly influence the probability of illness. Spearman rank correlation coefficients for the key variables were also determined with 100,000 iterations of a random run of the haemolytic ARRA scenario with the average washing variable.

## 3. Results

### 3.1. Occurrence and Concentration of Haemolytic, Ampicillin-Resistant, Penicillin G-Resistant and Tetracycline-Resistant Vibrio parahaemolyticus

The occurrence of haemolytic *V. parahaemolyticus* is summarized (Table 3). On the farm premises, the highest occurrences of haemolytic *V. parahaemolyticus* was observed in gills, followed by intestines and skin samples. Tetracycline-resistant *V. parahaemolyticus* isolates were completely absent in all sources sampled on the farm premises. Occurrences of haemolytic *V. parahaemolyticus* were higher in the retail premise compared to the farm premises. Occurrences of haemolytic *V. parahaemolyticus* were highest and similar for both gills and intestines, followed by skin samples on the retail premises. Overall, the occurrence trends of *V. parahaemolyticus* were highest in the haemolytic treatment, followed by ampicillin, penicillin G and, lastly, tetracycline treatment for both farm and retail premises.

The counts of haemolytic *V. parahaemolyticus* were summarised (Table 3). The highest levels of haemolytic *V. parahaemolyticus* were found in the intestines, followed by the gills and finally the skin. The described trend was observed in both farm and retail premises. Higher levels of haemolytic *V. parahaemolyticus* were observed in retail premises compared to farm premises within the same sample source. The highest levels of haemolytic *V. parahaemolyticus* counts from farm-derived water samples were found in haemolytic treatment, followed by ampicillin and, finally, penicillin G treatment. The occurrence percentage range of this study was similar to that of a study conducted by Siddique on a marine coastal farm rearing tilapia in southwest Bangladesh, while the concentration ranges of this study largely overlapped with that of a study performed by Ohno on horse mackerels landed in Niigata Prefecture, Japan, as well as that of another study by Tan on short mackerels obtained from retail premises [22,23,39].

### 3.2. Comparison of Risk Characterisation Outputs across Scenarios

The modelled occurrence and concentrations of AMR haemolytic *V. parahaemolyticus* for both the farm-to-home and retail-to-home scenarios were summarised (Table 4). All risk estimates, including the average probability of illness per serving (P_ill,serving_), probability of illness per person per year (P_ill,yearly_) and number of cases per annum (N_cases_) caused by AMR haemolytic *V. parahaemolyticus* infections under all different scenarios were summarized (Table 5). The haemolytic ARRA scenario predicted an average P_ill,serving_ of 19 per 100,000 servings (SE: 0.42 per 100,000) for the washing scenario, 112 per 100,000 servings (SE: 1.1 per 100,000) for the no-washing scenario and 29 per 100,000 servings (SE: 0.53 per 100,000) for the average washing scenario for the farm-to-home chain. An average P_ill,serving_ of 3 per 100,000 servings (SE: 0.069 per 100,000) for the washing scenario, 25 per 100,000 servings (SE: 0.38 per 100,000) for the no-washing scenario and 5 per 100,000 servings (SE: 0.084 per 100,000) for the average washing scenario were predicted for the retail-to-home chain. Within the haemolytic scenario, the comparison of the washing scenario and the no-washing scenario showed a reduction by 83% and 90% of the P_ill,serving_ for the farm-to-home chain and retail-to-home chain, respectively.

The haemolytic ARRA scenario was designated as the baseline for which other ARRA scenarios were compared against, with all other factors, the washing variable and chain variable remaining constant. The ratios of the other ARRA scenarios to the baseline scenario regarding the average P_ill,yearly_ were shown (Table 6). A much higher ratio values were observed for the retail-to-home chain as compared to the farm-to-home chain.

### 3.3. Sensitivity Analysis

The model inputs that have the greatest impacts on the variability on P_ill,serving_ were described for both the farm-to-home and retail-to-home chain scenarios (Figure 2. The cooking effect had the greatest impact on P_ill,serving_ variability for both scenarios, followed by the pre-harvest/retail start intestines and gills haemolytic *V. parahaemolyticus* concentrations and overall washing effect as the next top three inputs. The risk estimates were sensitive to a lesser extent to the fish, intestines and gills weight for both scenarios, as well as the harvest duration and serving size for the farm-to-home chain. All other model inputs, such as farm or retail prevalence, retail display duration and temperature, home transport and retail transport duration and temperature did not greatly influence the risk estimate results.

## 4. Discussion

### 4.1. Occurrence and Concentration Trends of Vibrio parahaemolyticus in Grey Mullet

It has been established that the skin, gills and intestines of finfish are prime locations for the colonisation of *Vibrio* spp. in the marine environment [21,22,23]. In this study, a lower occurrence of haemolytic *V. parahaemolyticus* (29%, 13/45) sampled at the farm premise was observed (Table 3). In contrast, the occurrence of haemolytic *V. parahaemolyticus* was higher when the grey mullet samples were transported to the hypermarket (60%, 27/45) (Table 3). In both premises, most of the haemolytic *V. parahaemolyticus* were mainly located in the gills and intestines of the grey mullet. Haemolytic *V. parahaemolyticus* concentrations were also lower in freshly harvested grey mullet from the marine fish farm (3.3 and 3.6 Log_10_CFU for gills and intestines, respectively) compared to grey mullet obtained from the hypermarket (3.4 and 3.8 Log_10_CFU for gills and intestines, respectively) (Table 3). A much greater amount of time would have elapsed during the transportation to the retail point, accounting for the growth of the foodborne pathogen as it progresses along the food chain and resulting in higher values of occurrences and concentrations observed. The higher occurrence and concentration of haemolytic *V. parahaemolyticus* in the intestines and gills might be attributed to the relatively larger surface-area-to-volume ratios of such organs, which also have a rich supply of blood that the pathogen can utilise for growth.

### 4.2. Comparison of ARRA Risk Estimates to Other Studies

The haemolytic ARRA scenario results, which did not consider AMR traits within the pathogen, were used for comparison with other studies. Within the haemolytic scenario, the P_ill_serving_ for the average washing effect for the full farm-to-home chain and partial retail-to-home chain was 2.9 × 10^−4^ and 4.5 × 10^−5^, respectively (Table 5). In comparison, a study by Iwahori reported a mean P_ill_serving_ of 1.4 × 10^−4^ for the worst-case scenario and 5.6 × 10^−6^ for the best-case scenario, while another study by Tan reported a mean P_ill_serving_ of 1.5 × 10^−9^ [28,32]. Overall, the risk estimates within this study are comparably similar to the study by Iwahori but are relatively higher by up to five magnitudes of order compared to the study by Tan [28,32]. The differences in P_ill_serving_ may be explained by how the cooking is modelled. The pathogen log reduction range by pan-frying cooking was specified at −1.1 to −7.0 in this study, based on cooking simulations performed by Ye, which is more conservative compared to the cooking log reduction range of −2.5 to −7.5 modelled by Tan, resulting in a relatively higher estimate of risk in this study [32,34]. Another reason for the relatively higher risk estimates could be due to the initial concentrations of *V. parahaemolyticus* pathogens in the fish. Temperature plays an important role in directly influencing the growth of haemolytic *V. parahaemolyticus* [40,41,42] (Boonyawantang et al., 2012; Desmarchelier, 1997; Dupray and Cormier, 1983). As Singapore is a tropical country with warm sea temperatures of around 26 °C–32 °C, this might explain the higher occurrence and concentration of *V. parahaemolyticus* in grey mullet fish.

Knowledge regarding sporadic gastroenteritis cases in Singapore is limited. Furthermore, cases of gastroenteritis caused by foodborne *V. parahaemolyticus* are not legally notifiable in Singapore; hence, there is a paucity of clinical data to perform model validation. However, in a study by Gurpreet, roughly 5% of the Malaysian population will experience acute diarrhoea cases annually, of which 3% of that will be attributed to *V. parahaemolyticus* [43,44]. The use of this surrogate data applied to Singapore’s context equates to 285,179 sporadic gastroenteritis cases, of which 8,556 cases are caused by this pathogen from the consumption of all types of fish and shellfish. Overall, 15 cases annually per population or 2.6 × 10^−1^ cases per 100,000 person was estimated from consumption of locally farmed contaminated grey mullets that were undercooked from the surrogate data. In this study, the model predicted an average of 148 and 57 cases annually with the 5th and 95th percentile being 0 to 319 cases for the farm-to-home chain and with the 5th and 97.5th percentile being 0 to 313 cases for the retail-to-home chain (Table 5). Therefore, the current models used have been validated as the number of cases obtained from the surrogate data falls within the range of cases predicted within the statistical model, showing that the models are able to robustly assess and compare different scenarios caused by changes in the model inputs.

### 4.3. Comparison among ARRAs

A comparison was made for risk estimates from the haemolytic ARRA between the farm-to-home chain and the retail-to-home chain. P_ill_yearly_ risk estimates were 2.6-fold higher for the average washing scenario and 3.1-fold higher for the washing scenario in the retail-to-home chain compared to the farm-to-home chain (Table 5). The opposite trend was observed when the risk estimates were 1.9-fold higher for the no-washing scenario in the farm-to-home chain compared to the retail-to-home chain (Table 5). Overall, the risk estimates concur with the occurrence and concentration findings of haemolytic *V. parahaemolyticus*, wherein higher values were reported at the retail premises compared to the farm premises, owing to increased time and temperature as the grey mullet moves through the food chain. Subsequently, this led to increased risk estimates for the retail-to-home chain compared to the farm-to-home chain. Such increased occurrence and concentration findings at the retail premise indicate the importance of time and temperature control in suppressing the growth of the pathogen within the food chain.

A comparison was also made in the haemolytic ARRA between the washing scenario and no-washing scenario for both the farm-to-home and retail-to-home chain. P_ill_yearly_ risk estimates were 3.6-fold higher in the no-washing scenario compared to the washing scenario for the farm-to-home chain and 5.9-fold higher for the retail-to-home chain (Table 5). The decrease in risk associated with the washing scenario indicates that this measure plays an important role in influencing risk estimates and subsequently, food safety.

Comparisons of the P_ill_yearly_ risk estimates across different ARRAs for the full farm-to-home chain and partial retail-to-home chain showed different trends. Within the farm-to-home chain, P_ill_yearly_ risk estimates for haemolytic ARRA are overall shown to be higher compared to the other two ARRAs, with ratios of 1.1 × 10^−2^, 3.4 × 10^−4^ for the ampicillin and penicillin G ARRAs, respectively, while tetracycline-resistant pathogen strains could not be detected (Table 6). The data show that, when the fish are freshly harvested, risk estimates from haemolytic AMR *V. parahaemolyticus* were relatively lower compared to the risk estimates from haemolytic non AMR *V. parahaemolyticus*. However, the trend is reversed for P_ill_yearly_ risk estimates in the retail-to-home chain, with a lower estimate for haemolytic non-AMR *V. parahaemolyticus* compared to the ampicillin- and penicillin G-resistant haemolytic *V. parahaemolyticus*, with ratios of 1.3 and 1.6 for the former and latter (Table 6). The risk estimate for haemolytic non-AMR *V. parahaemolyticus* was higher compared to tetracycline-resistant *V. parahaemolyticus* with a ratio of 4.2 × 10^−1^ (Table 6). These trends indicate that, for the farm-to-home chain, the gastroenteritis cases predicted would most likely be attributed to haemolytic non-AMR *V. parahaemolyticus* strains, while virtually all gastroenteritis cases predicted in the retail-to-home chain would be attributed to ampicillin- and penicillin G-resistant haemolytic strains, with possibly up to 42% of gastroenteritis cases caused by the tetracycline-resistant strain. Within the farm-to-home chain, the risk estimate was the highest for ampicillin-resistant strains, followed by penicillin G-resistant strains and was absent for tetracycline strains. In contrast, within the retail-to-home chain, risk estimates were the highest for penicillin G-resistant strains, followed by ampicillin-resistant strains and lastly for tetracycline strains. The results indicate that, overall, there is a greater risk of gastroenteritis caused by the consumption of grey mullet contaminated by ampicillin- or penicillin G-resistant *V. parahaemolyticus* compared to tetracycline-resistant *V. parahaemolyticus*.

### 4.4. Exposure Risk Estimates, Sensitivity Analysis of Intervention Measures and Limitations of Study

Based on the sensitivity analysis of the ARRAs for the full farm-to-home chain and partial retail-to-home chain, the key variables that influence the risk are as follows, in descending order, (1) the pan-frying cooking effect, followed by (2) the initial concentration of the *V. parahaemolyticus* pathogen in both the intestines and the gills of the grey mullet, and finally (3) the washing of the fish body cavity. Both the cooking and washing are related to consumer’s food handling and preparatory practices and are shown to be negatively correlated to risk, whereas the initial concentration of the pathogens is related to aquaculture farm’s rearing practices and is shown to be positively correlated to risk. From the consumer’s perspective, the risks are highest when the food is improperly handled or processed. Therefore, it is recommended that the fish body cavity is properly washed with clean water prior to cooking and to ensure that the fish is properly cooked prior to consumption to greatly reduce risk. From the aquaculture farmer’s perspective, good aquacultural practices or husbandry measures could be taken in order to reduce pathogen loads within the fish. In Singapore, this can be achieved by following the good aquaculture practices for fish farm guidelines set forth by SFA [45]. This can include improving the proper management of water quality within the farm; monitoring fish health, including vaccinations; and the prudent use of chemotherapeutants and proper feed management strategies. Other possible intervention measures include reducing time duration and temperature control within the food chain [29]. This can include reducing time for harvesting, retail display and transportation durations, although these measures do only have a marginal effect on reducing risk. While there were a limited number of samples collected, this study foremost aims to provide risk estimates pertaining to AMR in *V. parahaemolyticus* that are relevant to the local aquaculture industry and is not a prevalence study. Such estimates can then not only be used to contribute to AMR surveillance studies but can also be used by relevant stakeholders to make informed risk management decisions.

## 5. Conclusions

This is the first study that reported occurrences and concentrations of pathogenic AMR *V. parahaemolyticus* in grey mullet in Singapore. The findings of the study show that haemolytic AMR *V. parahaemolyticus* were detected in the skin, gills and intestines of grey mullet samples obtained from both farm and hypermarket premises. Occurrence and concentration of the AMR *Vibrio parahaemolyticus* in grey mullet were higher in the hypermarket compared to the farm. This study also assessed the risks imposed by the studied AMR haemolytic *V. parahaemolyticus* strains found in locally farmed grey mullets. The findings of this study show that, overall, the risk estimates were higher in the retail-to-home chain as compared to the farm-to-home chain. Furthermore, the risk of gastroenteritis caused by ampicillin- and penicillin G-resistant *V. parahaemolyticus* strains were higher compared to tetracycline-resistant strains. The sensitivity analysis also highlighted key model inputs that greatly influenced risk in the model, such as the cooking effect and the initial concentration of the pathogen within the fish, as well as the washing effect, allowing intervention measures to be crafted around these factors to mitigate risk.

One key limitation is that cross-contamination models are not considered in this study. Cross-contamination can play significant roles in altering the concentrations and occurrence of the pathogen, and it has been reported that up to 40 to 60% of foodborne illnesses are linked to surface cross-contamination [46], which can greatly influence risk estimates. Therefore, additional data is needed to understand the probability and extent of cross-contamination events within the risk framework, such as the handling and transportation of grey mullets to the retailer and retail display when all other seafood products from other sources are displayed together on fish ice beds. In addition, performing ARRAs requires large amounts of data, especially in the exposure assessment, wherein laboratory experiments need to be conducted, which can be time consuming and laborious, limiting sample sizes and increasing uncertainty within the model. However, even with such limitations, the generation of such risk estimates through computational modelling for the local aquaculture industry can be used by relevant stakeholders to make informed decisions regarding food safety practices.

In the future, whole-genome sequencing can be carried out on larger sample sizes of isolated *V. parahaemolyticus* isolates to understand the genotypic basis of their pathogenicity and AMR determinants, thus improving the exposure assessment and overall model accuracy [47]. This study can thus serve as a platform to provide risk data to relevant stakeholders, such as consumers of seafood, governmental bodies and aquaculture farmers, to make evidence-based risk management decisions. This can include education on proper food handling measures for consumers, better aquaculture rearing practices for farmers and policy planning for governmental bodies to improve overall food safety.

## Figures and Tables

**Figure 1 pathogens-12-00093-f001:**
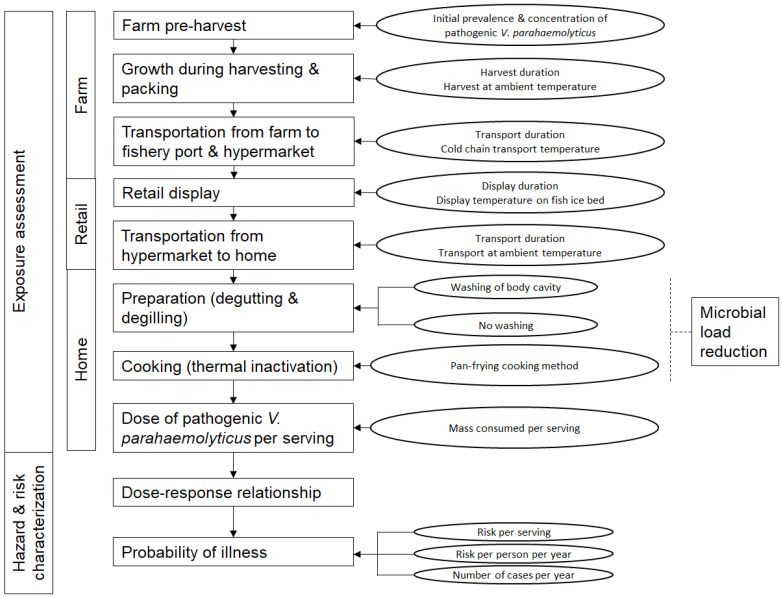
Quantitative risk evaluation model framework depicting the ARRA of haemolytic *V. parahaemolyticus* isolated from grey mullets.

**Figure 2 pathogens-12-00093-f002:**
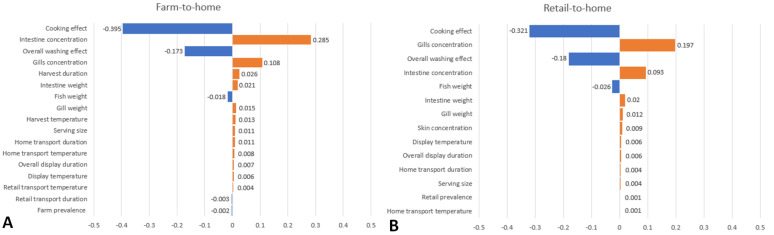
Tornado charts with model input variables influencing average P_ill,serving_ in haemolytic ARRA scenario using Spearman rank correlation coefficients. (**A**) Farm-to-home chain. (**B**) Retail-to-home chain.

**Table 1 pathogens-12-00093-t001:** All model input parameters for quantitative ARRA of haemolytic *V. parahaemolyticus* (*Vp*) in grey mullets.

Symbol	Description	Equation	References
Exposure assessment			
Growth rate equations			
$\sqrt{r}$	Growth rate in broth model (Log_10_/min)	$\frac{0.035634\left(T-278.5\right)\left[1-exp(0.3403\left(T-319.6\right)\right)]*\sqrt{\left(a_{w}-0.921\right)\left[1-exp(263.64\left(a_{w}-0.998\right)\right)]}}{\sqrt{ln(10)}}$	[27]
$R_{\mathrm{AD}}$	Growth rate adjustment (Log_10_/h)	$\frac{{\left(\sqrt{r}\right)}^{2}*60}{\mathrm{RiskTriang}\left(2,4,5\right)}$	[28,29]
Initial occurrence and concentration equations			
$P_{{\mathrm{patho}}_{\mathrm{farm}}}$	Occurrence of haemolytic *Vp*	RiskBeta (positives + 1, negatives + 1)	Author’s input
$Vp_{\mathrm{pre}\text{-}\mathrm{harvest}}$ ^#^	Total concentration of *Vp* in fish body (Log_10_/g)	$Log\left[\frac{\left(10^{Log\;\mathrm{CFU}/20\;{\mathrm{cm}}^{2}\;\mathrm{on}\;\mathrm{flesh}}*\left(\frac{S}{20\;{\mathrm{cm}}^{2}}\right)\right)+\left(10^{Log\;\mathrm{CFU}/g\;\mathrm{on}\;\mathrm{gills}}*\mathrm{gill}\;\mathrm{weight}\right)+\left(10^{Log\;\mathrm{CFU}/g\;\mathrm{on}\;\mathrm{intestines}}*\mathrm{intestine}\;\mathrm{weight}\right)}{\mathrm{Total}\;\mathrm{fish}\;\mathrm{weight}}\right]$	[28,30]
Growth during harvest			
$t_{\mathrm{harvest}}$	Harvest time (h)	RiskTriang (1, 1.5, 2)	Author’s input
$T_{\mathrm{harvest}}$	Harvest temperature (K)	RiskPert (299.05, 301.55, 305.45)	[31]
$Vp_{\mathrm{post}\text{-}\mathrm{harvest}}$	Concentration of *Vp* (Log_10_/g)	$Vp_{\mathrm{pre}\text{-}\mathrm{harvest}}+\left(R_{\mathrm{AD}}*t_{\mathrm{harvest}}\right)$	-
Growth during transport to retail			
$t_{F\to R}$	Transport time (h)	RiskUniform (13.5, 14.5)	Author’s input
$T_{F\to R}$	Transport temperature (K)	RiskPert (276.15, 279.15, 282.15)	[28]
$Vp_{\mathrm{retail}\;\mathrm{start}}$	Concentration of *Vp* (Log_10_/g)	$Vp_{\mathrm{post}\text{-}\mathrm{harvest}}+\left(R_{\mathrm{AD}}*t_{F\to R}\right)$	-
Growth during retail display			
$t_{\mathrm{retail}\text{-}90\%}$	Display time for majority of purchases (h)	RiskUniform (0.5, 3.5)	Author’s input
$t_{\mathrm{retail}\text{-}10\%}$	Display time for remaining purchases (h)	RiskPert (3.5, 5.5, 12.5)	Author’s input
$t_{\mathrm{overall}\;\mathrm{retail}}$	Overall display time (h)	$\mathrm{RiskDiscrete}(t_{\mathrm{retail}\text{-}90\%}:t_{\mathrm{retail}\text{-}10\%})$	Author’s input
$T_{\mathrm{retail}}$	Display temperature (K)	RiskPert (278.15, 283.15, 288.15)	[32]
$Vp_{\mathrm{retail}\;\mathrm{end}}$	Concentration of *Vp* (Log_10_/g)	$Vp_{\mathrm{retail}\;\mathrm{start}}+\left(R_{\mathrm{AD}}*t_{\mathrm{overall}\;\mathrm{retail}}\right)$	-
Growth during transport to home			
$t_{R\to H}$	Transport time (h)	RiskTriang (0.25, 0.5, 0.75)	[32]
$T_{R\to H}$	Transport temperature (K)	RiskPert (299.05, 301.55, 305.45)	[31]
$Vp_{\mathrm{home}}$	Concentration of *Vp* (Log_10_/g)	$Vp_{\mathrm{retail}\;\mathrm{end}}+\left(R_{\mathrm{AD}}*t_{\mathrm{overall}\;\mathrm{retail}}\right)$	-
Preparation and cooking			
${\mathrm{Prep}}_{\mathrm{wash}}$	Fish body cavity, washing scenario (Log_10_/g)	RiskNormal (−1.9921, 0.4545)	[33]
${\mathrm{Prep}}_{\mathrm{no}\;\mathrm{wash}}$	Fish body cavity, no washing scenario (Log_10_/g)	RiskNormal (−0.8449, 0.4897)	[33]
${\mathrm{Prep}}_{\mathrm{overall}}$	Overall washing scenario (Log_10_/g)	Prep_wash_—90% Prep_no wash_—10% ${\mathrm{Prep}}_{\mathrm{overall}}=\mathrm{RiskDiscrete}({\mathrm{Prep}}_{\mathrm{no}\;\mathrm{wash}}:{\mathrm{Prep}}_{\mathrm{no}\;\mathrm{wash}})$	Author’s input
${\mathrm{Cook}}_{\mathrm{pan}\;\mathrm{fry}}$	Heat inactivation (Log_10_/g)	RiskUniform (−1.1, −7)	[34]
$Vp_{{\mathrm{prep}}_{i}}$ ^&^	Concentration of *Vp* (Log_10_/g)	$Vp_{\mathrm{home}}+{\mathrm{Prep}}_{i}$	-
$Vp_{{\mathrm{dose}}_{i}}$	Concentration of *Vp* (Log_10_/g)	$Vp_{{\mathrm{prep}}_{i}}+{\mathrm{Cook}}_{\mathrm{pan}\;\mathrm{fry}}$	-
Serv	Serving Size (grams)	RiskTriang (0, 61.2, 197.3)	[35]
d	Dose (CFU)	$10^{Vp_{{\mathrm{dose}}_{i}}}*Serv$	-
Hazard characterisation			
BP	Beta Poisson dose-response	$1-{\left(1+\frac{d}{\beta }\right)}^{-\alpha }$	[29]
$P_{{\mathrm{patho}}_{F\to R}}$ ^@^	Occurrence change from farm to retail	$P_{{\mathrm{patho}}_{\mathrm{farm}}}+\left(1-P_{{\mathrm{patho}}_{\mathrm{farm}}}\right)*\left(P_{{\mathrm{patho}}_{\mathrm{retail}}}-P_{{\mathrm{patho}}_{\mathrm{farm}}}\right)$	-
$P_{\mathrm{ill},\mathrm{serving}}$	Probability of illness per serving	$BP*P_{{\mathrm{patho}}_{i}}$ ^@^	-
Risk characterisation			
$Pol_{\mathrm{Singapore}}$	Singapore’s population	5,703,569	[36]
$Pol_{\mathrm{fish}}$	Population proportion consuming finfish	0.92	[37]
$Pol_{\mathrm{grey}\;\mathrm{mullet}}$	Population proportion consuming grey mullet	$\frac{150}{87,306}=1.718\times 10^{-3}$	Author’s input
$Pol_{\mathrm{exposed}}$	Exposed population	$Pol_{\mathrm{Singapore}}*Pol_{\mathrm{fish}}*Pol_{\mathrm{grey}\;\mathrm{mullet}}=9015$	-
n	Number of meals per week	RiskNormal (10.37, 7.586)	[37]
$P_{\mathrm{ill},\mathrm{yearly}}$	Probability of illness per person per year	$1-{\left(1-P_{\mathrm{ill},\mathrm{serving}}\right)}^{\left(n*52\right)}$	-
$N_{\mathrm{cases}}$	Cases per year	$P_{\mathrm{ill},\mathrm{yearly}}*Pol_{\mathrm{exposed}}$	-

^#^ Values of *Vp*_pre-harvest_ were based on the four different ARRA scenarios—haemolytic, ampicillin, penicillin G and tetracycline; ^&^
i = Wash, no wash or overall preparation scenarios. ^@^
$P_{{\mathrm{patho}}_{F\to R}}$ or $P_{{\mathrm{patho}}_{\mathrm{Retail}}}$ scenarios.

**Table 2 pathogens-12-00093-t002:** Beta-Poisson dose response MLEs of α and β and the corresponding probability weight.

FDA Model	α	β	Probability Weight
1	1.47 × 10^6^	3.53 × 10^14^	3.40 × 10^−4^
2	1.26 × 10^7^	7.20 × 10^14^	4.12 × 10^−3^
3	6.37 × 10^2^	1.65 × 10^10^	2.06 × 10^−2^
4	3.58 × 10^1^	5.42 × 10^8^	5.49 × 10^−2^
5	2.08 × 10^1^	1.99 × 10^8^	8.23 × 10^−2^
6	1.49 × 10^1^	8.78 × 10^7^	6.58 × 10^−2^
7	1.06 × 10^1^	2.99 × 10^7^	2.20 × 10^−2^
8	3.89	2.28 × 10^8^	6.90 × 10^−4^
9	1.31	2.93 × 10^7^	8.23 × 10^−3^
10	5.20 × 10^−1^	3.61 × 10^6^	4.12 × 10^−2^
11	4.70 × 10^−1^	1.50 × 10^6^	1.10 × 10^−1^
12	6.00 × 10^−1^	1.31 × 10^6^	1.65 × 10^−1^
13	1.00	1.80 × 10^6^	1.32 × 10^−1^
14	8.59	1.30 × 10^7^	4.39 × 10^−2^
15	1.50 × 10^−1^	2.33 × 10^5^	3.40 × 10^−4^
16	1.90 × 10^−1^	2.29 × 10^5^	4.12 × 10^−3^
17	2.50 × 10^−1^	2.36 × 10^5^	2.06 × 10^−2^
18	3.20 × 10^−1^	2.57 × 10^5^	5.49 × 10^−2^
19	4.30 × 10^−1^	3.04 × 10^5^	8.23 × 10^−2^
20	6.90 × 10^−1^	4.34 × 10^5^	6.58 × 10^−2^
21	6.92	4.49 × 10^6^	2.20 × 10^−2^

**Table 3 pathogens-12-00093-t003:** Occurrence (%) and concentration (S.E.) of haemolytic *V. parahaemolyticus* isolates in grey mullet and farm water samples from a coastal marine farm and a hypermarket in Singapore. AMP-R: ampicillin-resistant, PENG-R: penicillin G-resistant; tetracycline-resistant. Percentages are calculated with denominator being sample total/water sample.

VP	Marine Coastal Farm				Hypermarket				Farm Water
	Sample Total	Gill	Skin	Intestine	Sample Total	Gill	Skin	Intestine	
Occurrence—haemolytic	45	8/15 (18)	0/15 (0)	5/15 (11)	45	12/15 (27)	4/15 (9)	11/15 (24)	2/6 (33)
Mean Concentration —haemolytic	-	3.3 (0.39)	0 (0)	3.6 (0.74)	-	3.4 (0.38)	2.3 (0.23)	3.8 (0.24)	3.0 (0.035)
Occurrence—AMP-R	45	1/15 (2)	0/15 (0)	0/15 (0)	45	12/15 (27)	4/15 (9)	0/15 (0)	2/6 (33)
Mean Concentration—AMP-R	-	2.3 (0)	0 (0)	0 (0)	-	3.7 (0.33)	2.3 (0.23)	3.7 (0.32)	2.7 (0.41)
Occurrence—PENG-R	45	2/15 (4)	0/15 (0)	0/15 (0)	45	9/15 (20)	7/15 (16)	10/15 (22)	2/6 (33)
Mean Concentration—PENG-R	-	2.2 (0.15)	0 (0)	0 (0)	-	3.0 (0.21)	1.9 (0.12)	4.1 (0.43)	2.0 (0)
Occurrence—TET-R	45	0/15 (0)	0/15 (0)	0/15 (0)	45	12/15 (27)	0/15 (0)	9/15 (20)	0/6 (0)
Mean Concentration—TET-R	-	0 (0)	0 (0)	0 (0)	-	2.1 (0.20)	0 (0)	3.0 (0.49)	0 (0)

**Table 4 pathogens-12-00093-t004:** Occurrence and concentration changes of haemolytic *V. parahaemolyticus* within the farm-to-home and retail-to-home chain scenarios. Occurrence and concentration data were determined using Monte Carlo simulations for 20 runs and 100,000 iterations per run. Percentile values in brackets represent the 5th and 95th, respectively.

			Farm-to-Home				Retail-to-Home			
			Haemolytic	Ampicillin	Penicillin G	Tetracycline	Haemolytic	Ampicillin	Penicillin G	Tetracycline
	Occurrence (Farm/Retail)		2.9 × 10^−1^ (1.9 × 10^−1^, 4.1 × 10^−1^)	4.3 × 10^−2^ (7.8 × 10^−3^, 9.9 × 10^−2^)	6.4 × 10^−2^ (1.8 × 10^−2^, 1.3 × 10^−1^)	N.A.	6.0 × 10^−1^ (4.8 × 10^−1^, 7.1 × 10^−1^)	5.7 × 10^−1^ (4.6 × 10^−1^, 6.9 × 10^−1^)	5.7 × 10^−1^ (4.6 × 10^−1^, 6.9 × 10^−1^)	4.7 × 10^−1^ (3.5 × 10^−1^, 5.9 × 10^−1^)
Concentra-tion (LogCFU/g)	Farm	Pre-harvest	2.4 (6.5 × 10^−1^, 4.5)	6.8 × 10^−1^ (0, 1.9)	4.5 × 10^−1^ (0, 8.6 × 10^−1^)	N.A.	N.A.	N.A.	N.A.	N.A.
		Post-harvest	2.9 (1.1, 4.9)	1.2 (3.7 × 10^−1^, 2.4)	9.2 × 10^−1^ (4.6 × 10^−1^, 1.4)	N.A.	N.A.	N.A.	N.A.	N.A.
	Retail	Retail-start	2.9 (1.1, 5.0)	1.2 (3.8 × 10^−1^, 2.4)	9.3 × 10^−1^ (4.7 × 10^−1^, 1.4)	N.A.	2.7 (1.5, 4.1)	2.8 (1.6, 4.2)	2.7 (1.3, 4.5)	1.6 (1.0 × 10^−1^, 3.6)
		Retail-end	3.0 (1.2, 5.0)	1.2 (4.1 × 10^−1^, 2.4)	9.7 × 10^−1^ (5.1 × 10^−1^, 1.4)	N.A.	2.7 (1.6, 4.1)	2.8 (1.6, 4.3)	2.7 (1.4, 4.6)	1.7 (1.4 × 10^−1^, 3.7)
	Home	Home	3.1 (1.3, 5.1)	1.35 (5.6 × 10^−1^, 2.6)	1.1 (6.6 × 10^−1^, 1.6)	N.A.	2.9 (1.7, 4.3)	3.0 (1.8, 4.4)	2.8 (1.5, 4.7)	1.8 (3.0 × 10^−1^, 3.8)
	Preparation	Average Washing	1.3 (0, 3.5)	1.6 × 10^−1^ (0, 1.0)	4.9 × 10^−2^ (0, 4.0 × 10^−1^)	N.A.	1.0 (0, 2.7)	1.2 (0, 2.9)	1.1 (0, 3.0)	4.5 × 10^−1^ (0, 2.1)
		Washing	1.2 (0, 3.3)	1.1 × 10^−1^ (0, 7.8 × 10^−1^)	1.2 × 10^−2^ (0, 1.5 × 10^−2^)	N.A.	9.2 × 10^−1^ (0, 2.4)	1.1 (0, 2.6)	9.5 × 10^−1^ (0, 2.9)	3.8 × 10^−1^ (0, 2.0)
		No washing	2.3 (2.9 × 10^−1^, 4.4)	6.2 × 10^−1^ (0, 1.9)	3.8 × 10^−1^ (0, 1.2)	N.A.	2.0 (5.9 × 10^−1^, 3.6)	2.1 (6.6 × 10^−1^, 3.8)	2.0 (4.0 × 10^−1^, 4.0)	1.1 (0, 3.1)
	Cooking	Average Washing	8.9 × 10^−2^ (0, 6.8 × 10^−1^)	1.3 × 10^−3^ (0, 0)	3.4 × 10^−5^ (0, 0)	N.A.	3.7 × 10^−2^ (0, 7.6 × 10^−2^)	4.7 × 10^−2^ (0, 2.5 × 10^−1^)	5.5 × 10^−2^ (0, 2.7 × 10^−1^)	1.7 × 10^−2^ (0, 0)
		Washing	7.3 × 10^−2^ (0, 5.1 × 10^−1^)	4.5 × 10^−4^ (0, 0)	2.5 × 10^−7^ (0, 0)	N.A.	2.6 × 10^−2^ (0, 0)	3.4 × 10^−2^ (0, 7.6 × 10^−2^)	4.3 × 10^−2^ (0, 9.8 × 10^−2^)	1.2 × 10^−2^ (0, 0)
		No washing	2.4 × 10^−1^ (0, 1.7)	9.3 × 10^−3^ (0, 0)	3.7 × 10^−4^ (0, 0)	N.A.	1.4 × 10^−1^ (0, 1.0)	1.6 × 10^−1^ (0, 1.2)	1.6 × 10^−1^ (0, 1.3)	5.7 × 10^−2^ (0, 2.9 × 10^−1^)

**Table 5 pathogens-12-00093-t005:** Comparison of risk estimates across all scenarios. Average P_ill,serving_, P_ill,yearly_ and N_cases_ were determined using Monte Carlo simulations for 20 runs and 100,000 iterations per run. Percentile values in brackets represent the 5th and 95th, respectively.

		Farm-to-Home			Retail-to-Home		
		P_ill,serving_	P_ill,yearly_	N_cases_	P_ill,serving_	P_ill,yearly_	N_cases_
Haemolytic	Average Washing	2.9 × 10^−4^ (0, 9.0 × 10^−5^)	1.6 × 10^−2^ (0, 3.5 × 10^−2^)	1.5 × 10^2^ (0, 3.2 × 10^2^)	4.5 × 10^−5^ (0, 8.4 × 10^−6^)	6.3 × 10^−3^ (0, 3.6 × 10^−4^)	5.7 × 10^1^ (0, 3.3)
	Washing	1.9 × 10^−4^ (0, 5.7 × 10^−5^)	1.3 × 10^−2^ (0, 2.1 × 10^−2^)	1.2 × 10^2^ (0, 1.9 × 10^2^)	2.4 × 10^−5^ (0, 0)	4.3 × 10^−3^ (0, 0)	3.8 × 10^1^ (0, 0)
	No washing	1.1 × 10^−3^ (0, 8.6 × 10^−4^)	4.7 × 10^−2^ (0, 3.2 × 10^−1^)	4.2 × 10^2^ (0, 2.8 × 10^3^)	2.5 × 10^−4^ (0, 2.6 × 10^−4^)	2.5 × 10^−2^ (0, 1.1 × 10^−1^)	2.3 × 10^2^ (0, 1.0 × 10^3^)
Haemolytic and AMP-R	Average Washing	4.5 × 10^−7^ (0, 0)	1.8 × 10^−4^ (0, 0)	1.6 (0, 0)	5.2 × 10^−5^ (0, 2.7 × 10^−5^)	8.0 × 10^−3^ (0, 7.9 × 10^−3^)	7.2 × 10^1^ (0, 7.1 × 10^1^)
	Washing	1.3 × 10^−7^ (0, 0)	5.9 × 10^−5^ (0, 0)	5.3 × 10^−1^ (0, 0)	2.6 × 10^−5^ (0, 8.3 × 10^−6^)	5.4 × 10^−3^ (0, 4.9 × 10^−4^)	4.9 × 10^1^ (0, 4.4)
	No washing	3.4 × 10^−6^ (0, 0)	1.3 × 10^−3^ (0, 0)	1.1 × 10^1^ (0, 0)	2.8 × 10^−4^ (0, 3.7 × 10^−4^)	3.1 × 10^−2^ (0, 1.6 × 10^−1^)	2.8 × 10^2^ (0, 1.4 × 10^3^)
Haemolytic and PENG-R	Average Washing	1.0 × 10^−8^ (0, 0)	5.6 × 10^−6^ (0, 0)	5.0 × 10^−2^ (0, 0)	1.2 × 10^−4^ (0, 2.8 × 10^−5^)	1.0 × 10^−2^ (0, 7.7 × 10^−3^)	9.1 × 10^1^ (0, 7.0 × 10^1^)
	Washing	7.2 × 10^−11^ (0, 0)	5.3 × 10^−8^ (0, 0)	4.8 × 10^−4^ (0, 0)	7.4 × 10^−5^ (0, 9.8 × 10^−6^)	7.7 × 10^−3^ (0, 6.7 × 10^−4^)	6.9 × 10^1^ (0, 6.1)
	No washing	1.0 × 10^−7^ (0, 0)	5.4 × 10^−5^ (0, 0)	4.9 × 10^−1^ (0, 0)	5.7 × 10^−4^ (0, 3.8 × 10^−4^)	3.2 × 10^−2^ (0, 1.5 × 10^−1^)	2.9 × 10^2^ (0, 1.4 × 10^3^)
Haemolytic and TET-R	Average Washing	N.A.	N.A.	N.A.	2.3 × 10^−5^ (0, 0)	2.6 × 10^−3^ (0, 0)	2.4 × 10^1^ (0, 0)
	Washing	N.A.	N.A.	N.A.	1.3 × 10^−5^ (0, 0)	1.9 × 10^−3^ (0, 0)	1.7 × 10^1^ (0, 0)
	No washing	N.A.	N.A.	N.A.	1.1 × 10^−4^ (0, 2.5 × 10^−5^)	9.4 × 10^−3^ (0, 7.0 × 10^−3^)	8.5 × 10^1^ (0, 6.3 × 10^1^)

**Table 6 pathogens-12-00093-t006:** Ratios of the average P_ill,yearly_ of the ARRA scenarios to the average P_ill,yearly_ of the haemolytic scenario.

		Farm-to-Home	Retail-to-Home
Haemolytic	Average washing	1	1
	Washing	1	1
	No washing	1	1
Ampicillin	Average washing	1.1 × 10^−2^	1.3
	Washing	4.5 × 10^−3^	1.3
	No washing	2.7 × 10^−2^	1.2
Penicillin G	Average washing	3.4 × 10^−4^	1.6
	Washing	4.1 × 10^−6^	1.8
	No washing	1.2 × 10^−3^	1.3
Tetracycline	Average washing	NA	4.2 × 10^−1^
	Washing	NA	4.4 × 10^−1^
	No washing	NA	3.8 × 10^−1^

## Data Availability

https://www.dropbox.com/scl/fo/npiq159jmrycfv97tr91q/h?dl=0&rlkey=t61rebtteg0bv1zalc5np674e.

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
