# Peer review of "Quantitative Risk Evaluation of Antimicrobial-Resistant Vibrio parahaemolyticus Isolated from Farmed Grey Mullets in Singapore"

_pathogens, 2023, doi:10.3390/pathogens12010093_

Round 1
Reviewer 1 Report
The authors created their study thesis, results and conclusions on the investigation of only 30 fish (l. 136), and subsequenlty colleceted 27 isolates (supplementary data), recuperated between November 2019 and January 2020.
With such a low number of investigated samples and isolated strains you must not do statistics as conducted within the study.
As there are 35,000 human cases only in the USA per year (l. 47), it should be very easy to sample a higher amount of fish and collect a higher number of isolates.
Please include more samples within your study and resubmit the interesting study results.
Thank you!
Reviewer 2 Report
In this manuscript, the authors evaluated the risk of antimicrobial resistant Vibrio parahaemolyticus from farmed grey mullets in Singapore. The purpose of this study is clear and the content is rich. This manuscript could be acceptable for the publication in the journal of Pathogens after the minor revisions.
1. In keywords: what dose “@Risk” mean?
2. Introduction should be added some information about the definition of antimicrobial-resistant bacteria.
3. In sample processing, whether the size of the fish collected each time is consistent?
4. How to ensure the freshness of each sample is consistent?
5. According to ampicillin, penicillin G and tetracycline resistance, the strain was determined to be antimicrobial resistant (AMR) strain. Please provide the determination basis. And these three antibiotics are not widely used in aquaculture, why did you choose these three drugs?
Round 2
Reviewer 1 Report
Thank you for your reviesed version.
Please revise the Abstract in line 22: two out of six samples must not declared as 33%, as this is statistically not possible! You may write "one third".
